# Identifying the Producer and Grade of Matcha Tea through Three-Dimensional Fluorescence Spectroscopy Analysis and Distance Discrimination

**DOI:** 10.3390/foods12193614

**Published:** 2023-09-28

**Authors:** Yue Xu, Xiangyang Zhou, Wenjuan Lei

**Affiliations:** 1College of Tea Science, Guizhou University, Guiyang 550025, China; gs.xuy21@gzu.edu.cn; 2College of Resources and Environmental Engineering, Guizhou University, Guiyang 550025, China; xyzhou6@gzu.edu.cn; 3Key Laboratory of Karst Geological Resources and Environment, Ministry of Education, Guizhou University, Guiyang 550025, China

**Keywords:** matcha tea, discriminant analysis, producer and grade, three-dimensional fluorescence spectroscopy

## Abstract

The three-dimensional fluorescence spectroscopy features the advantage of obtaining emission spectra at different excitation wavelengths and providing more detailed information. This study established a simple method to discriminate both the producer and grade of matcha tea by coupling three-dimensional fluorescence spectroscopy analysis and distance discrimination. The matcha tea was extracted three times and three-dimensional fluorescence spectroscopies of these tea infusions were scanned; then, the dimension of three-dimensional fluorescence spectroscopies was reduced by the integration at three specific areas showing local peaks of fluorescence intensity, and a series of vectors were constructed based on a combination of integrated vectors of the three tea infusions; finally, four distances were used to discriminate the producer and grade of matcha tea, and two discriminative patterns were compared. The results indicated that proper vector construction, appropriate discriminative distance, and correct steps are three key factors to ensure the high accuracy of the discrimination. The vector based on the three-dimensional fluorescence spectroscopy of all three tea infusions resulted in a higher accuracy than those only based on spectroscopy of one or two tea infusions, and the first tea infusion was more sensitive than the other tea infusion. The Mahalanobis distance had a higher accuracy that was up to 100% when the vector is appropriate, while the other three distances were about 60–90%. The two-step discriminative pattern, identifying the producer first and the grade second, showed a higher accuracy and a smaller uncertainty than the one-step pattern of identifying both directly. These key conclusions above help discriminate the producer and grade of matcha in a quick, accurate, and green method through three-dimensional fluorescence spectroscopy, as well as in quality inspections and identifying the critical parameters of the producing process.

## 1. Introduction

Matcha is a type of green tea (*Camellia sinensis*) with special cultivation and processing methods, which require that tea leaves be covered and shaded from excessive direct sunlight for at least twenty days before harvesting [1,2,3]. These processes are used to produce higher amounts of amino acids and bioactive compounds, such as increasing chlorophyll content to confer matcha its lush green color and to retain the antioxidants in the leaves to preserve matcha’s intense flavor [4]. Many studies have reported that matcha tea contains a relatively high content of catechins, caffeine, rutin, quercetin, vitamin C, chlorophyll, and theanine compared with other green teas [4,5,6,7,8]. These ingredients promote health, possessing anticarcinogenic effects [9,10], anti-inflammatory effects [11], and cardioprotective effects [7], as well has having the potential to regulate carbohydrate metabolism [12], improve cognitive function, and prevent of neurodegenerative disorders [4]. As a result, matcha tea is rapidly becoming a new alternative drink or a food supplement for more people around the world [13,14,15].

Correspondingly, the judgment of quality and the identification of authentic and fake matcha tea are widespread consumer concerns. Usually, the producer and grades are the two key points lending themselves to the identification of matcha tea, and human panel testing and chemical analysis are the most elementary and commonly used conventional methods for analyzing matcha tea [16]. However, the abovementioned methods are time-consuming, laborious, and non-green [16,17,18]. Spectroscopy analysis is increasingly widely used to identify and quantify the composition of tea. Ultraviolet spectroscopy is capable of testing organic matter containing double bonds and has been used to differentiate the varieties of tea plants [18,19], but it usually requires a pretreatment that can destroy the structures. Visible spectroscopy can be used to obtain the color, shape, and fractal features, while failing to explore the microscopic composition. Near infrared (NIR) spectroscopy features the advantages of fast speed, easy or no pretreatment, acceptable accuracy, and environmental friendliness [20], and has been widely used for qualitative identification of the key components [21,22,23] and the varieties and grades of tea [24,25,26,27]. To obtain a higher testing accuracy, hyperspectral images have also been employed for the abovementioned discriminations [28,29,30,31]. Moreover, three-dimensional fluorescence spectroscopy analysis can obtain emission spectra for different excitation wavelengths, which can provide more detailed information and are more sensitive [32]. Then, these experimental data are widely mined by multivariate statistical analysis methods. The key variables are identified and their dimensions are reduced through principal component analysis [17,33,34,35,36] and linear discriminant analysis [17,18] before being clarified by support vector machines [18]. The correlation between the spectroscopy signals and contents of specific components are regressed by partial least squares class-modeling [1,13,34,35,37], parallel factor analysis [35], and self-weighted alternative trilinear decomposition [34] and artificial neural networks [1,17,18,38]. The accuracy is higher than 81%, with some reaching up to 100%.

These spectroscopy-based methods feature the merits of fast speed, easy or no pretreatment, and an acceptable accuracy, though they have been scarcely used in the discriminations of matcha. Several studies established a good correlation between the key components and particle size of matcha and the specific intervals of visible and near-infrared spectroscopy [13,39,40], implying a potential applicability to the others. Three-dimensional fluorescence spectroscopy (excitation-emission matrices, EEMs) is widely used to analyze the molecular structure, environment factors, and functions of food products due to the advantages of fast speed, less sampling, good selectivity, high sensitivity, and good reproducibility [34,35,38,41,42]. Unfortunately, it has little application in the discrimination of matcha, such as the variety, producer, and grade.

The objective of this study, therefore, was to test the applicability of discriminating matcha by three-dimensional fluorescence spectroscopy analysis and discriminative methods and identify the appropriate pattern based on the accuracy evaluation. The results are helpful in quality inspections and critical parameter identification of matcha tea in a quick, accurate, and green method.

## 2. Materials and Methods

### 2.1. Matcha Tea Collection and Preparation

All matcha tea samples were purchased through local markets or online shopping platforms and were from different provinces (Guizhou, Anhui, and Henan) in China. The particle size of the three matcha tea samples follows the national standard of China (GB/T 34778-2017) [43] with the D60 ≤ 18 μm. Moreover, each Matcha tea covers all the grades labeled by the producer from the best of level 1 to the worst of 5 or 6. Each grade of these matchas includes 10 random samples, and 170 matcha tea samples in total were used to test the discrimination. These matcha samples were packaged and sub-packed in hermetic bags and stored at −20 °C to maintain their quality until all experiments were completed. The producer, grade, productive year, and the number of samples are shown in Table 1.

Tea infusions were prepared as follows [44,45]: 0.50 g of matcha tea sample was mixed with 50 mL of distilled water at 80 °C, then water bath heat preservation was conducted at 80 °C for 30 min, followed by centrifugation and collection of the supernatant. The supernatant was diluted ten times to ensure a good performance when scanning the fluorescence spectroscopy, which requires the maximum intensity of fluorescence spectroscopy to be less than 8000 during a test. Every sample was extracted three times consecutively following the method above, and the supernatants of tea fusions were collected.

Apparatus and Conditions: Three-dimensional fluorescence spectroscopy (excitation-emission matrices, EEMs) measurements were performed directly on matcha tea infusion at room temperature using a 1.00 cm quartz cell on a Lengguang 97pro spectrophotometer (Shanghai, China). EEM fluorescence spectra were obtained by recording the emission spectra from 250 to 800 nm (with 2 nm intervals) corresponding to excitation wavelengths ranging between 250 and 800 nm (with 10 nm intervals), and the scan rate was 48,000 nm/min. Each scan comprised 65 emission and 275 excitation wavelengths, making a total of 17,875 spectral points per sample matrix. Rayleigh and Raman scattering points per sample matrices were roughly corrected by subtracting the average response matrix of blank solutions.

### 2.2. Methodologies

#### 2.2.1. Three-Dimensional Fluorescence Spectroscopy

The contour maps of exemplary three-dimensional fluorescence spectroscopy of these matcha infusions are shown in Figure 1. It can be seen there were one to three characteristic peaks at different emission and excitation wavelengths, denoted by Em and Ex for short respectively. Previous studies have revealed that the fluorescence spectroscopy at Ex/Em = 320–410/390–520 nm, Ex/Em = 410/500 nm, and Ex/Em = 380–480/650–700 nm represents flavonoids, tea polyphenols, and chlorophyll a, respectively [44]. The fluorescence characteristics of these tea infusions exhibited obvious differences with the change of producers, grades, and times of extraction, as shown in Figure 1.

#### 2.2.2. Dimensionality Reduction of Three-Dimensional Fluorescence Spectroscopy

(1)Selection of the integral domains

Based on the exemplary three-dimensional fluorescence spectroscopy of matcha tea in Figure 1, we selected three integral regions that exhibit local peaks of fluorescence intensity that can reflect the key components of tea [44]. The first region is 320 nm ≤ Ex ≤ 410 nm, and 390 nm ≤ Em ≤ 520 nm; the second region is 410 nm ≤ Ex ≤ 500 nm and 490 nm ≤ Em ≤ 560 nm, and the third region is 380 nm ≤ Ex ≤ 480 nm and 650 nm ≤ Em ≤ 700 nm, respectively. Then, the dimension of three-dimensional fluorescence spectroscopy was reduced by the integration of regional fluorescence intensity, *I_RFI_*. The integration of a specific region showing a local peak is calculated below:(1)$$I_{FRI}=\sum_{\mathrm{ex}}\sum_{em}I(\lambda_{ex}\lambda_{em})\Delta \lambda_{ex}\Delta \lambda_{em}$$
where *I_FRI_* is integration of regional fluorescence intensity at a specific domain showing the local peak; $\Delta \lambda_{ex}$ and $\Delta \lambda_{em}$ are excitation and emission wavelength intervals, respectively; and $I(\lambda_{ex}\lambda_{em})$ is fluorescence intensity.

(2)Vectors of the integrated three-dimensional fluorescence spectroscopy

After the integration, the three-dimensional fluorescence spectroscopy of a matcha tea infusion is represented by a vector with three elements, as denoted by Formula (2).
(2)$$V_{Int1}=[I_{11},\;I_{12},\;I_{13}]$$
where *V_Int_*_1_ is the vector calculated by the integration of three-dimensional fluorescence spectroscopy, *I*_11_, *I*_21_, and *I*_31_ is the integrated value of the first tea infusion based on the three regions displaying local peaks.

When matcha tea is extracted two or three times, a six-dimensional or nine-dimensional vector is constructed based on the three-dimensional fluorescence spectroscopy of these tea infusions:(3)$$V_{Int2}=[I_{11},\;I_{12},\;I_{13},\;I_{21},\;I_{22},\;I_{23}]$$
(4)$$V_{Int3}=[I_{11},\;I_{12},\;I_{13},\;I_{21},\;I_{22},\;I_{23},\;I_{31},\;I_{32},\;I_{33}]$$
where *V_Int_*_2_ is the vector calculated by the integration of three-dimensional fluorescence spectroscopy of the first and the second tea infusion; *V_Int_*_3_ is calculated by the integration of all the three tea infusions; *I*_21_, *I*_22_, and *I*_23_ is the integrated value of the second tea infusion, and *I*_31_, *I*_32_, and *I*_33_ is the integrated value of the third tea infusion based on the three regions, respectively.

#### 2.2.3. Discrimination Based on Different Distances

##### Different Distances

(1)Mahalanobis distances

Assuming the population *μ* with the dimension *n*, the mean vector is denoted as
(5)$$\mu =[\mu_{1},\;\mu_{2}\;,\;\ldots \;,\;\mu_{n}]$$
and the sample *X* is denoted as
(6)$$X=[X_{1},\;X_{2}\;,\;\ldots \;,\;X_{n}]$$

The Mahalanobis distances between the sample and the population are calculated below.
(7)$$d_{Mah}=(X-\mu )C^{-1}{\left(X-\mu \right)}^{-1}$$
where *d_Mah_* is the Manalanobis distances and *C* is the covariance matrix.

The advantage of the Manalanobis distances is independent of the measured scale of the variables. The Manalanobis distance depends on the covariance matrix of training samples, the calculation fails when the size of the covariance matrix is less than the length of the vector. Thus, several other distances are used to complement.

(2)Three other distances

Three distances are independent from the covariance matrix of the population and the total samples are given below, including Euclidean distance, Manhattan distance, and Chebychev distance.

①Euclidean distances

The Euclidean distances, *d_Ecu_*, between the two vectors *X* and *Y* with dimension *n* are calculated below:(8)$$d_{Ecu}=\sqrt{{\left(X_{1}-Y_{1}\right)}^{2}+{\left(X_{2}-Y_{2}\right)}^{2}+,\;\ldots \;,\;+{\left(X_{n}-Y_{n}\right)}^{2}}$$

②Manhattan distances

The Manhattan distances, *d_Man_*, also called the city block distance, are calculated below:(9)$$d_{Man}=\sum_{i=1}^{n}\left|X_{i}-Y_{i}\right|$$

③Chebychev distances

The Chebychev distances, *d_Che_*, determined by the maximum distance of the corresponding elements between the two vectors, are shown below:(10)$$d_{\mathrm{Che}}=\max\left|X_{i}-Y_{i}\right|$$

(3)Transformation of the distances

In order to keep the consistency of the coordinate system from 0 to 1, the relative values of the distances above were used for the discrimination.
(11)$$d_{ri}=\frac{d_{i}-d_{\min}}{d_{\max}-d_{\min}}$$
where *d_ri_* is the relative distance of the tested sample, *d_i_* is the distance of the testing sample, and *d_max_* and *d_min_* are the maximum and minimum distances of all the testing samples, respectively.

##### Discrimination

The general discrimination is based on the minimum distances between the testing sample and training samples.
(12)$$D_{res}=\min(d_{ri})$$
where *D_res_* is the result of discrimination, *d_ri_* is the distance to the population *i*.

### 2.3. Construction of Technical System

#### 2.3.1. Technical Route Diagram

The technical route diagram of discrimination is given in Figure 2, and generally, it includes five main steps.
(1)Tea infusion extraction and three-dimensional fluorescence spectroscopy scanning. Matcha tea was extracted three times consecutively and the diluted tea infusion is scanned by three-dimensional fluorescence spectroscopy;(2)Reduce the dimension of EMMs by the integration. Then, the three-dimensional fluorescence spectroscopy of a tea fusion was integrated at three specific regions, and a tea fusion can provide seven reconstructed vectors based on the random combination of the three extractions;(3)Characteristic vectors training of the population. Following that, the trained characteristic vectors covering the three matchas and all of their grades were obtained by the average value of the training samples. In this study, 170 samples were used to test the discrimination. As for each one, the other 169 samples from the three manufacturers with 17 grades were trained;(4)Distance calculation and discrimination. After that, the four distances mentioned above were calculated between the vectors of the testing sample and the training population. The samples were discriminated by the minimum distance between them;(5)Accuracy evaluation and parameter optimization. The results calculated from different distances and vectors displayed a series of accuracies, from which the appropriate vector and distances were also identified.

#### 2.3.2. Discriminative Patterns

This study aimed at identifying both the producer or manufacturer and the grade of different matchas. Usually, the number of the trained samples of matcha is several times larger than that of the samples known as both producer and grade, e.g., each matcha includes five or six grades in this study. Hence, two discriminative patterns were tested: two-step and one-step pattern. The former identifies the producer or producing area firstly based on the training samples of all grades of a matcha and then identifies its grade based on the characteristic vectors of the identified samples only. The latter identifies both the producing area and grade together by comparing the distances of training samples of all grades of different matchas. Both patterns are illustrated in Figure 3.

### 2.4. The Accuracy Test

#### 2.4.1. One-Step Pattern

The accuracy of the one-step pattern discriminant is as follows.
(13)$$A_{1}=\frac{n}{N}\times 100\%$$
where A_s1_ is the accuracy of the one-step pattern discriminant, *N* is the number of the total tested number, and n is the number of samples recognized correctly.

#### 2.4.2. Two-Step Pattern

Assuming A_1_ and A_2_ are the accuracies of the producing area discriminant and grade discriminant with the premise of a known producer or producing area, respectively. The two steps are independent of each other; hence, the accuracy of identifying both the producer and grade of the matcha tea can be considered a conditional probability.
(14)$$P(A|B)=\frac{P(\mathrm{AB})}{P(B)}\Leftrightarrow P(\mathrm{AB})=P(A|B)P(B)$$

Here, the accuracy of the first and second steps, A_1_ and A_2_, corresponds to P(B) and P(A|B) in Formula (14) respectively, and the accuracy of identifying both, denoted as *A*_S2_, corresponding to P(AB).
(15)$$A_{12}=A_{1}A_{2}$$

## 3. Results and Discussions

### 3.1. Two-Step Discriminative Pattern

#### 3.1.1. Identification of the Producing Area

(1)Spatial distributions of tested samples in distance space

The three-dimensional coordinate system is constructed based on the distances to the trained samples of three matchas, and the spatial distributions of all the tested samples are shown in Figure 4, which, generally, showed two typical characteristics. On one hand, the samples of matcha from the same producer displayed a good spatial continuity and were located in different areas of the coordinate system, although the grades of these samples are different. This implies the applicability of discriminating the varieties of matcha tea based on the methodologies mentioned above. On the other hand, the spatial distributing patterns also displayed some differences among the results from four distances. Generally, the results calculated from the Mahalanobis distance are more prone to being concentrated, while showing a larger spread based on the other three distances. The different spatial distribution is crucial to identify the appropriate distance to the recognition.

(2)The accuracy of producer identification

The correct rates of identifying the producing area based on the discrimination from four distances are shown in Table 2. It can be observed that the accuracy varied substantially with the change of discriminative distances and construction of vectors based on three-dimensional fluorescence spectroscopy of different tea infusions. As for the former, Mahalanobis distance performed better and the accuracy was 5–30% higher than the discriminations from the other three distances. As for the latter, the vectors showing a higher dimension had a better accuracy, and the first tea fusion was more sensitive than the second and the third, manifesting in accuracies of 98.24%, 70.59%, and 90.59%, respectively.

#### 3.1.2. Grade Discrimination

The second step, grade discrimination of matcha tea, is based on the premise that the producer is known, and the discriminative process is the same as the former; the results are shown in Table 3. The results revealed that the accuracy of grade discrimination was similar to that of the producing area. Totally, the accuracy of Mahalanobis distance-based discrimination is higher than the three others, and the average accuracy of 170 testing samples showed a minimum of 96.67% based on three-dimensional fluorescence spectroscopy of the first tea infusion, and was up to 100% when the integrated vector was based on at least the first two tea infusions.

#### 3.1.3. Accuracy of Discriminating Both Producing Area and Grade

Based on Formula (13), the accuracy of identifying both producer and grade were obtained. The results shown in Table 4 indicate that the Mahalanobis distance-based discriminant showed a high accuracy of more than 98% when the vector is established based on the three-dimensional fluorescence spectroscopies of the first and second tea infusions at least and can be up to more than 99% when all the three tea infusions are considered. However, the results based on the other three distances showed a much lower accuracy of less than 80%. Therefore, the Mahalanobis distance-based two-step pattern performed well in identifying both the producing areas and grades of matcha tea through at least two tea infusions.

Besides, two aspects should be noticed based on the results above. The first is the accuracy varies among the producers. There is a higher accuracy to matcha B and C discriminated by all distances while a lower accuracy to matcha A. The second is the accuracy from three distances except Mahalanobis do not show an obvious increase trend with increasing dimensions of integrated vector. This is not consistent with the regular expectation, implying a more sensitive discrimination based on Mahalanobis distance. Hence, the satisfactory result can be obtained through the two-step discriminative pattern by coupling three-dimensional fluorescence spectroscopy analysis and Mahalanobis distance, and the accuracy be up to 100%, although there is probably some randomness in the samples.

### 3.2. One-Step Discriminative Pattern

The one-step discriminative pattern was also carried out. In this case, the training sample only contained the samples with the same producer and grade. The results based on the four discriminative distances are shown in Table 5. It was found that the accuracy resulted from Euclidean distance, Manhattan distance, and Chebychev distance displayed a close level to those of known produces in Section 3.1.2. Additionally, the accuracy displayed an obvious difference among the three matchas, ranging from 38.33% to 78% for matcha A, 36% to 92% for matcha B, and from 48% to 98.33% for matcha C, respectively.

However, the results calculated from the Mahalanobis distance showed quite a different pattern and exhibited values close to zero, implying the failure of the one-step discriminative pattern using the Mahalanobis distance. The failure can be mainly attributed to the small number of training samples that resulted in the covariance matrix is not fullyy ranked. During the one-step discriminative pattern, the number of training samples was 9, implying the rank of the covariance matrix may be less than 9 if some elements are linearly correlated. In that case, it will lead to the failure to calculate Mahalanobis distance. Apart from that, the correlations among samples from the same producer with different grades are not considered in a one-step discriminative pattern, and the loss of information may reduce the accuracy.

Therefore, the one-step discriminative pattern is prone to failure based on Mahalanobis distance. The other three distances can provide a replacement, but the main challenge is the accuracy being 15% to 25% lower than the two-step discriminative pattern.

## 4. Conclusions

Based on the results of different vectors, discriminative distances, and discriminative patterns, the main conclusions are as follows. (1)The vector based on the integration of three-dimensional fluorescence spectroscopy of matcha tea infusion plays an important role in determining the accuracy of the discriminant. In total, the vectors calculated from the three-dimensional fluorescence spectroscopy of the first tea infusion exhibited an accuracy about 25–50% higher than the second and third tea infusion-based vectors. The vector based on more tea fusions had a higher accuracy;(2)The Mahalanobis distance had a higher accuracy that was up to 100% when the vector was appropriate, while the other three distances were about 60–90%. However, the Mahalanobis distance was challenged by the small number of training samples which is prone to lead to the matrix not being full in the calculation;(3)The two-step discriminative pattern, identifying the producer first and the grade second, showed a higher accuracy and a smaller uncertainty than the one-step discriminative pattern. This is because the correlations among samples from the same producer with different grades are not considered in the one-step discriminative pattern.

These key conclusions above help discriminate the producing area or producer and grade of matcha in a quick, accurate, and green method through three-dimensional fluorescence spectroscopy, as well as in identifying the critical parameters of producing the matcha.

## Figures and Tables

**Figure 1 foods-12-03614-f001:**
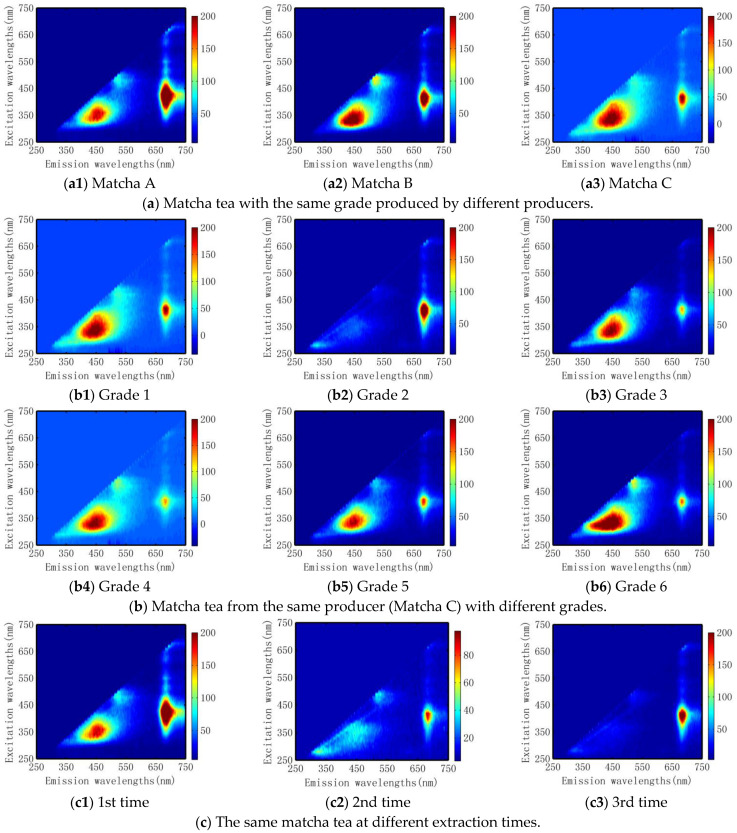
Contour maps of exemplary three-dimensional fluorescence spectroscopies of different matcha tea infusions. Subfigures (**a1**–**a3**) show the differences of excitation-emission matrices from the tea infusion of different matcha tea extracted the first time; (**b1**–**b6**) show the differences among the Matcha tea from the same produced area with different grades; and (**c1**–**c3**) show the differences among the tea fusions extracted from first to the third time of matcha tea with the same producing area and grade.

**Figure 2 foods-12-03614-f002:**
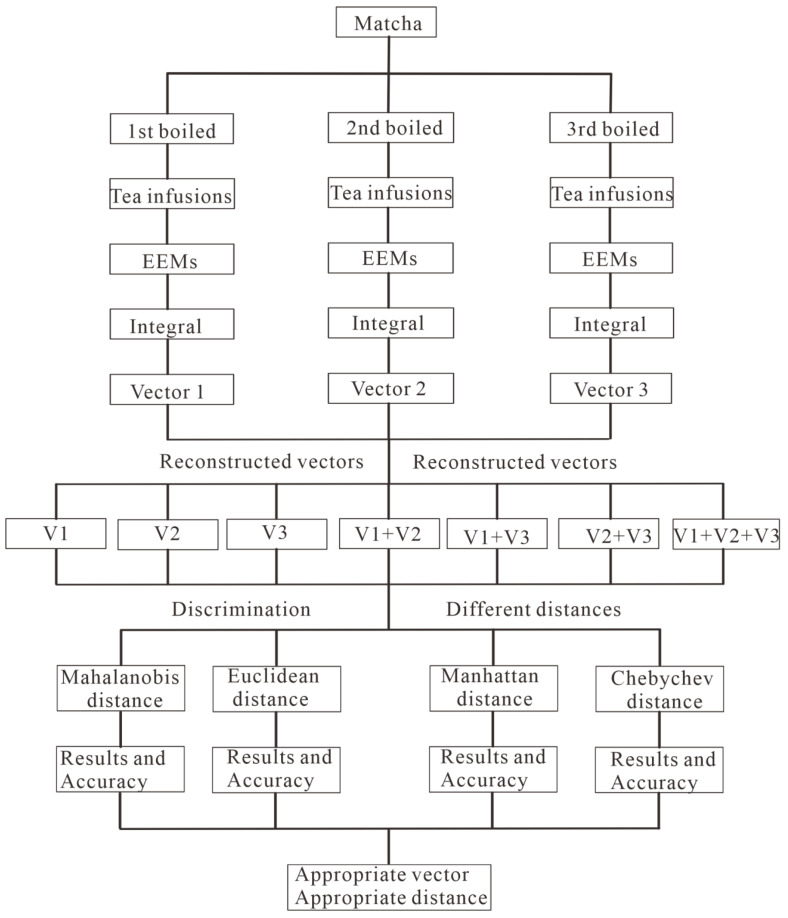
Technical route diagram of discrimination through three-dimensional fluorescence spectroscopy method and pattern recognition techniques. EEMs are excitation-emission matrices; V1 is the vector integrated from the EEMs of the first tea infusion in three specific areas: (1) 320 nm ≤ Ex ≤ 410 nm, 390 nm ≤ Em ≤ 520 nm; (2) 410 nm ≤ Ex ≤ 500 nm, and 490 nm ≤ Em ≤ 560 nm and (3) 380 nm ≤ Ex ≤ 480 nm, and 650 nm ≤ Em ≤ 700 nm, respectively. V2 and V3 are obtained from the integration of the EEMs of the second and third tea infusions.

**Figure 3 foods-12-03614-f003:**
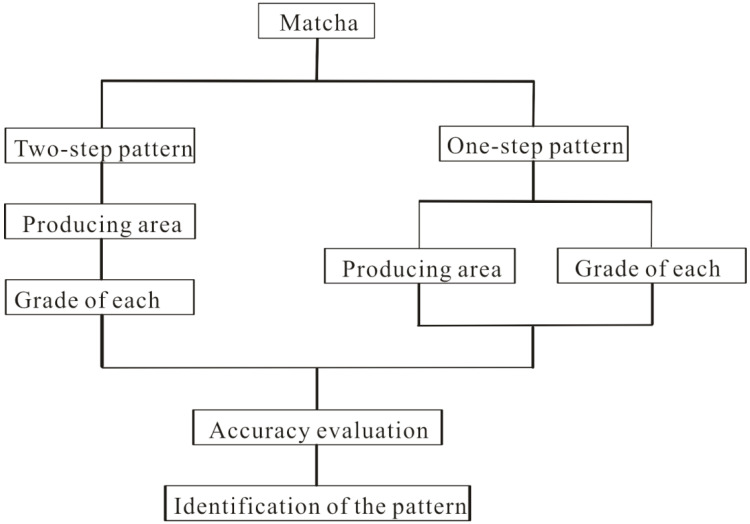
Illustration of different discriminative patterns: two steps vs. one step. The one-step discriminative pattern identifies both producer and grade simultaneously. The two-step pattern identifies the producer first based on the minimum distance between the testing and training samples and then identifies the grade of samples with the same producers.

**Figure 4 foods-12-03614-f004:**
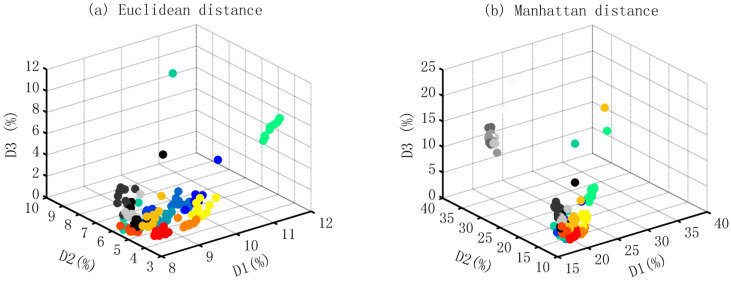
Spatial distributions of the test samples in the three-dimensional space. Note: The gray (ranging from black to white), autumn (from yellow to red) and winter (from green to blue) filled cycles represent the mocha from producing areas A, B and C respectively. The grades were labeled with different colors. D1, D2, and D3 are the distances to the trained sample of matcha A, B, and C respectively; the distance is the relative value ranging from 0 to 1, which is calculated by the linear transformation: (*d* − *d_min_*)/(*d_max_* − *d_min_*).

**Table 1 foods-12-03614-t001:** Summary of the test matcha tea samples.

Types	Place of Production	Year	Grade	Parallel Samples	Number of Samples	Times to Tea Infusion	Number of Tea Infusions
A	Guizhou	2021	6	10	60	3	180
B	Anhui	2021	5	10	50	3	150
C	Henan	2021	6	10	60	3	180
In total	-	-	17	-	170	-	510

**Table 2 foods-12-03614-t002:** The accuracy of producer discrimination for matcha tea based on different distances.

Matcha	Distance	Tea Infusion						
		P1	P2	P3	P1 + P2	P1 + P3	P2 + P3	P1 + P2 + P3
A	Euclidean	90.00%	85.00%	83.33%	95.00%	95.00%	83.33%	98.33%
	Manhattan/city block	96.67%	85.00%	83.33%	98.33%	98.33%	90.00%	98.33%
	Chebychev	86.67%	85.00%	78.33%	88.33%	93.33%	78.33%	93.33%
	Mahalanobis	100.00%	66.67%	91.67%	100.00%	100.00%	96.67%	100.00%
B	Euclidean	86.00%	74.00%	58.00%	90.00%	84.00%	74.00%	86.00%
	Manhattan/city block	88.00%	68.00%	62.00%	94.00%	86.00%	76.00%	90.00%
	Chebychev	76.00%	74.00%	60.00%	78.00%	76.00%	70.00%	78.00%
	Mahalanobis	94.00%	52.00%	98.00%	94.00%	100.00%	96.00%	98.00%
C	Euclidean	91.67%	66.67%	66.67%	76.67%	80.00%	66.67%	71.67%
	Manhattan/city block	93.33%	66.67%	66.67%	81.67%	85.00%	66.67%	80.00%
	Chebychev	85.00%	65.00%	66.67%	70.00%	81.67%	66.67%	70.00%
	Mahalanobis	100.00%	90.00%	83.33%	100.00%	100.00%	90.00%	100.00%
Mean	Euclidean	89.41%	75.29%	70.00%	87.06%	86.47%	74.71%	85.29%
	Manhattan/city block	92.94%	73.53%	71.18%	91.18%	90.00%	77.65%	89.41%
	Chebychev	82.94%	74.71%	68.82%	78.82%	84.12%	71.76%	80.59%
	Mahalanobis	98.24%	70.59%	90.59%	98.24%	100.00%	94.12%	99.41%

Note: P1, P2, and P3 mean the infusion abstracted the first, second, and third time from the distilled water at 80 °C, and the accuracy was calculated from 170 testing samples.

**Table 3 foods-12-03614-t003:** The accuracy of grade discrimination of three different matcha teas using four typical distances with the premise of a known producing area.

Matcha	Distance	Tea Infusion						
		P1	P2	P3	P1 + P2	P1 + P3	P2 + P3	P1 + P2 + P3
A	Euclidean	61.67%	53.33%	46.67%	65.00%	76.67%	55.00%	75.00%
	Manhattan/city block	63.33%	58.33%	48.33%	66.67%	76.67%	60.00%	81.67%
	Chebychev	56.67%	53.33%	46.67%	58.33%	71.67%	48.33%	61.67%
	Mahalanobis	96.67%	76.67%	80.00%	100.00%	100.00%	90.00%	100.00%
B	Euclidean	94.00%	52.00%	46.00%	96.00%	84.00%	62.00%	86.00%
	Manhattan/city block	92.00%	46.00%	52.00%	92.00%	82.00%	62.00%	78.00%
	Chebychev	92.00%	50.00%	44.00%	96.00%	86.00%	50.00%	86.00%
	Mahalanobis	100.00%	88.00%	70.00%	100.00%	100.00%	100.00%	100.00%
C	Euclidean	96.67%	75.00%	78.33%	100.00%	100.00%	78.33%	100.00%
	Manhattan/city block	98.33%	80.00%	78.33%	100.00%	98.33%	76.67%	100.00%
	Chebychev	86.67%	73.33%	78.33%	95.00%	100.00%	75.00%	95.00%
	Mahalanobis	96.67%	91.67%	86.67%	100.00%	100.00%	100.00%	100.00%
Mean	Euclidean	83.53%	60.59%	57.65%	86.47%	87.06%	65.29%	87.06%
	Manhattan/city block	84.12%	62.35%	60.00%	85.88%	85.88%	66.47%	87.06%
	Chebychev	77.65%	59.41%	57.06%	82.35%	85.88%	58.24%	80.59%
	Mahalanobis	97.65%	85.29%	79.41%	100.00%	100.00%	96.47%	100.00%

Note: P1, P2, and P3 mean the infusion abstracted the first, second, and third time from the distilled water at 80 °C, and the accuracy was calculated from 170 testing samples.

**Table 4 foods-12-03614-t004:** The accuracy of a two-step discriminative pattern based on four different distances.

Matcha	Distance	Tea Infusion						
		P1	P2	P3	P1 + P2	P1 + P3	P2 + P3	P1 + P2 + P3
A	Euclidean	55.50%	45.33%	38.89%	61.75%	72.84%	45.83%	73.75%
	Manhattan/city block	61.22%	49.58%	40.27%	65.56%	75.39%	54.00%	80.31%
	Chebychev	49.12%	45.33%	36.56%	51.52%	66.89%	37.86%	57.56%
	Mahalanobis	96.67%	51.12%	73.34%	100.00%	100.00%	87.00%	100.00%
B	Euclidean	80.84%	38.48%	26.68%	86.40%	70.56%	45.88%	73.96%
	Manhattan/city block	80.96%	31.28%	32.24%	86.48%	70.52%	47.12%	70.20%
	Chebychev	69.92%	37.00%	26.40%	74.88%	65.36%	35.00%	67.08%
	Mahalanobis	94.00%	45.76%	68.60%	94.00%	100.00%	96.00%	98.00%
C	Euclidean	88.62%	50.00%	52.22%	76.67%	80.00%	52.22%	71.67%
	Manhattan/city block	91.77%	53.34%	52.22%	81.67%	83.58%	51.12%	80.00%
	Chebychev	73.67%	47.66%	52.22%	66.50%	81.67%	50.00%	66.50%
	Mahalanobis	96.67%	82.50%	72.22%	100.00%	100.00%	90.00%	100.00%
Mean	Euclidean	74.68%	45.62%	40.36%	75.28%	75.28%	48.78%	74.25%
	Manhattan/city block	78.18%	45.85%	42.71%	78.31%	77.29%	51.61%	77.84%
	Chebychev	64.40%	44.39%	39.27%	64.91%	72.24%	41.79%	64.95%
	Mahalanobis	95.93%	60.21%	71.94%	98.24%	100.00%	90.80%	99.41%

Note: P1, P2, and P3 mean the infusion abstracted the first, second, and third time from the distilled water at 80 °C, and the accuracy was calculated from 170 testing samples.

**Table 5 foods-12-03614-t005:** Accuracy of the one-step discriminative pattern based on four different distances.

Matcha	Distance	Tea Infusion						
		P1	P2	P3	P1 + P2	P1 + P3	P2 + P3	P1 + P2 + P3
A	Euclidean	60.00%	41.67%	41.67%	63.33%	75.00%	48.33%	73.33%
	Manhattan/city block	60.00%	48.33%	43.33%	65.00%	75.00%	58.33%	78.33%
	Chebychev	55.00%	38.33%	40.00%	56.67%	70.00%	41.67%	60.00%
	Mahalanobis	0.00%	0.00%	0.00%	0.00%	0.00%	0.00%	0.00%
B	Euclidean	92.00%	40.00%	38.00%	94.00%	84.00%	54.00%	86.00%
	Manhattan/city block	92.00%	34.00%	46.00%	90.00%	82.00%	56.00%	76.00%
	Chebychev	86.00%	38.00%	36.00%	92.00%	82.00%	40.00%	82.00%
	Mahalanobis	20.00%	20.00%	20.00%	20.00%	20.00%	20.00%	20.00%
C	Euclidean	91.67%	46.67%	71.67%	100.00%	91.67%	68.33%	98.33%
	Manhattan/city block	95.00%	53.33%	75.00%	96.67%	91.67%	73.33%	98.33%
	Chebychev	83.33%	48.33%	73.33%	91.67%	90.00%	68.33%	90.00%
	Mahalanobis	0.00%	0.00%	0.00%	0.00%	0.00%	0.00%	0.00%
Mean	Euclidean	80.59%	42.94%	51.18%	85.29%	83.53%	57.06%	85.88%
	Manhattan/city block	81.76%	45.88%	55.29%	83.53%	82.94%	62.94%	84.71%
	Chebychev	74.12%	41.76%	50.59%	79.41%	80.59%	50.59%	77.06%
	Mahalanobis	5.88%	5.88%	5.88%	5.88%	5.88%	5.88%	5.88%

Note: P1, P2, and P3 means the infusion abstracted the first, second, and third time from the distilled water at 80 °C, and the accuracy was calculated from 170 testing samples.

## Data Availability

The data used to support the findings of this study are available by the corresponding author upon request.

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
