# Peer review of "Identifying the Producer and Grade of Matcha Tea through Three-Dimensional Fluorescence Spectroscopy Analysis and Distance Discrimination"

_foods, 2023, doi:10.3390/foods12193614_

Round 1

Reviewer 1 Report

The authors examine Matcha tea to distinguish among different growth locations. Samples (tea infusions) are characterized using three-dimensional fluorescence spectroscopy, able to reveal the presence of chemical components in samples, the presence of which can be related to the overall quality.

1)     Line 58: The term “computer vision” is very confusing. The authors should use the term “hyperspectral images”.

2)     Lines 62-63: “self-weight alternative trilinear decomposition” or “self-weighted”?

3)     Lines 58-63: Discriminant techniques provide logic rules to assign samples to different groups. Such a method as, for instance, PCA is not a discriminant technique. Please rephrase the section considering this remark.

4)     Section 2.1: What exactly do you mean by quality levels?

5)     Line 83: “It can be seen, a total of 170 matcha tea samples are selected for analysis.” – from where can it be seen?

6)     Line 120: I suggest rephrasing the sentence: “Three consecutive extractions of the same Matcha tea sample.”.

7)     I wonder if three consecutive extractions are necessary. Differences in the overall quality and differences related to growth location must be observed already for the first tea infusion. Please compare results based on data obtained from single extraction and sum of three extractions and conclude if it has a substantial effect.

8)     Line 121: These are not “typical” patterns but “exemplary” EEMs.

9)     Section 2.2.2.: The authors did not explain why such spectral regions were selected.

10) Section 2.2.3.1.: The Mahalanobis distance can always be computed in the space of selected principal components. They are orthogonal, and thus it is possible to calculate the inverse of the covariance matrix.

11) Please use vector-matrix notation in all equations to avoid confusion.

12) Why do the authors use three different distances and not LDA or other pattern recognition techniques? A more interesting approach is to avoid integration, being to a large extent subjective, and construct classification models using complete EEM signals.

13) The authors examine samples from three growing regions. Using discriminant methods (e.g., LDA, different distances), it is possible to discriminate samples only if the number of groups is finite. Therefore, if the goal is to trace the geographical origin of samples, soft classification rules must be constructed (e.g., SIMCA). The discriminant methods are useful to reveal differences among groups of samples and identify relevant variables.

14) Line 207: “comparing with the trained samples” – what do you mean by “trained samples”? Representative samples from a given group are used to construct a model, and they are called either “a model set” or “training samples”.

15) Figure 2: I encourage the authors to modify the concept of the experiment and limit consideration to single extraction.

16) If four different distances are computed, how is the conclusion made?

17) This study concerns two discriminant problems – geographical origin and quality grade. I do not understand the motivation behind the “two-step pattern” and I do not find it useful to perform two discriminant tasks at once (see section 3.1.3.).

18) The authors should carefully revise the terminology they introduce. For instance, “two-step pattern of recognition” or “two-step pattern” is neither informative nor correct.

19) The models must be validated using an independent set of samples. How do the authors select samples to model and test set? As far as I understood, results in Table 1 are obtained only for training samples. In addition to accuracy, the authors must report sensitivity and specificity for model and test samples. Otherwise, models have no value.

20) Reference section: [5] and [7] – correct spelling of authors names.

21) Keywords – it is unnecessary to repeat the same words from the title in the keywords list.

Dear Editor,

Enclosed, please find my comments.

Kind regards,
Michal Daszykowski

Author Response

 Dear Review,

Thank you very much for reviewing our manuscript. Your questions and suggestions are helpful in improving this manuscript. Here, please check attached file for the responses.

Best regards

Reviewer 2 Report

1. In the introduction, what is the meaning of rapid instruments? Those mentioned methods are rapid in data acquisition. However, some methods like UV-Vis and fluorescence spectroscopy need laborious sample preparation. The authors should mention the merit and demerit of the previous methods proportionally.

2. Several attempts had been reported on the quality determination of matcha using NIR spectroscopy on powder both using benchtop and portable spectrometers. No need for sample preparation, fast and non-destructive. Please mention and formulate again the reason why it is needed to use 3-D fluorescence spectroscopy. NIR spectroscopy is also cheaper than 3-D fluorescence spectroscopy.

The authors may look at the following reports:

https://doi.org/10.1016/j.foodchem.2021.129372

https://doi.org/10.1016/j.foodchem.2023.136185

https://doi.org/10.1016/j.jfca.2022.104868

3. In Materials and methods

- What are the grades used in this study? What kind of grades? in tea products, there are several unique labels for tea grades. The authors may mention it in Table 1.

- What is the particle size of the tea samples? all same? What size? is there any difference in particle size of A, B, and C tea samples?

- Why dilution? is the spectral data saturated? dilution also highly influences the quality of fluorescence intensity. Why 10 times dilution?

- Figure 1. It is usually to plot the EEM with an x-axis for emission and a y-axis for excitation. 

4. In Results and discussions

- The label for Cityblock distance is replaced by Manhattan distance.

- Figure 4. I guess the authors use all samples (A, B, C) and all grades to generate Figure 4. The label is only for A, B, and C. Please add the label for grades. Please provide the value of D1, D2, and D3 in percent. Provide the total explained variance used to discriminate the tea samples. Using all distance methods, it is difficult to separate between B and C tea samples. Why? 

- How the accuracy is calculated? the authors separate the samples into training and prediction samples. So please show the number of correct and incorrect samples, not only the accuracy. Please revise Tables 2, 3, 4, and 5.

Author Response

Dear reviewer,
Thank you for spending time on reviewing our manuscript. Your comments and suggestions help us to improve the literature review, methodologies and results and discussion. Please check the attached file for the responses.

Best regards.

Round 2

Reviewer 1 Report

Dear Authors,

I am satisfied with the provided answers and introduced modifications.

Kind regards!

Reviewer 2 Report

The revision is acceptable.